# Multi-Method Quantification of Acetyl-Coenzyme A and Further Acyl-Coenzyme A Species in Normal and Ischemic Rat Liver

**DOI:** 10.3390/ijms241914957

**Published:** 2023-10-06

**Authors:** Malgorzata Tokarska-Schlattner, Nour Zeaiter, Valérie Cunin, Stéphane Attia, Cécile Meunier, Laurence Kay, Amel Achouri, Edwige Hiriart-Bryant, Karine Couturier, Cindy Tellier, Abderrafek El Harras, Bénédicte Elena-Herrmann, Saadi Khochbin, Audrey Le Gouellec, Uwe Schlattner

**Affiliations:** 1University Grenoble Alpes, Inserm U1055, Laboratory of Fundamental and Applied Bioenergetics (LBFA), 38058 Grenoble, France; nour.zeaiter@univ-grenoble-alpes.fr (N.Z.); stephane.attia@univ-grenoble-alpes.fr (S.A.); laurence.kay@univ-grenoble-alpes.fr (L.K.); amel.achouri1@univ-grenoble-alpes.fr (A.A.); edwige.hiriart-bryant@univ-grenoble-alpes.fr (E.H.-B.); karine.couturier@univ-grenoble-alpes.fr (K.C.); cindy.tellier@univ-grenoble-alpes.fr (C.T.); 2University Grenoble Alpes, CNRS UMR 5525, Laboratory TIMC—Translational Microbiology, Evolution, Engineering (TREE), Service de Biochimie, Biologie Moléculaire et Toxicologie Environnementale, CHU Grenoble-Alpes, 38058 Grenoble, France; vcunin@chu-grenoble.fr (V.C.); cmeunier@chu-grenoble.fr (C.M.); alegouellec@chu-grenoble.fr (A.L.G.); 3University Grenoble Alpes, Inserm U1209 and CNRS UMR5309, Institute for Advanced Biosciences (IAB), 38058 Grenoble, France; abderrafek.eh@gmail.com (A.E.H.); benedicte.elena@univ-grenoble-alpes.fr (B.E.-H.); saadi.khochbin@univ-grenoble-alpes.fr (S.K.); 4Institut Universitaire de France (IUF), 75231 Paris, France

**Keywords:** acetyl-CoA, acyl-CoA, fluorometric assay, HPLC, liver ischemia, mass spectrometry, metabolomics, NMR, spectrophotometric assay, succinyl-CoA

## Abstract

Thioesters of coenzyme A (CoA) carrying different acyl chains (acyl-CoAs) are central intermediates of many metabolic pathways and donor molecules for protein lysine acylation. Acyl-CoA species largely differ in terms of cellular concentrations and physico-chemical properties, rendering their analysis challenging. Here, we compare several approaches to quantify cellular acyl-CoA concentrations in normal and ischemic rat liver, using HPLC and LC-MS/MS for multi-acyl-CoA analysis, as well as NMR, fluorimetric and spectrophotometric techniques for the quantification of acetyl-CoAs. In particular, we describe a simple LC-MS/MS protocol that is suitable for the relative quantification of short and medium-chain acyl-CoA species. We show that ischemia induces specific changes in the short-chain acyl-CoA relative concentrations, while mild ischemia (1–2 min), although reducing succinyl-CoA, has little effects on acetyl-CoA, and even increases some acyl-CoA species upstream of the tricarboxylic acid cycle. In contrast, advanced ischemia (5–6 min) also reduces acetyl-CoA levels. Our approach provides the keys to accessing the acyl-CoA metabolome for a more in-depth analysis of metabolism, protein acylation and epigenetics.

## 1. Introduction

Acyl-CoAs represent the activated, energy-rich form of long, medium and short-chain fatty acids found in nutrients and along many metabolic pathways. Linked by a thioester bond to coenzyme A, they play central roles as metabolic intermediates in the breakdown of nutrients, as building blocks in anabolic pathways and as precursors of protein lysine acylation [1,2]. As such, they participate in the metabolism of proteins, carbohydrates, ketone bodies and lipids, as well as in cell signaling and epigenetic regulations. Most acyl-CoAs are primarily synthesized in mitochondria, although some synthesis could also occur in cytosol and nuclei [2]. The most abundant and central cellular acyl-CoA species is the short-chain acetyl-CoA (ace-CoA) [3]. It is synthesized in mitochondria by the catabolism of fatty acids (beta-oxidation), carbohydrates (pyruvate dehydrogenase) or branched-chain amino acids, before entering the tricarboxylic acid (TCA) cycle. Citrate exported from the TCA cycle into the cytosol is the primary source for cytosolic acetyl-CoA synthesis by ATP citrate lyase (ACLY). In addition, cytosolic and nuclear acetyl-CoA can be derived from ethanol (following the successive action of alcohol dehydrogenase and aldehyde dehydrogenase) and acetate (which can be directly used by acetyl-CoA synthetase short-chain 2, ACSS2). Other major cellular short-chain acyl-CoAs are succinyl-CoA (succ-CoA), a TCA cycle intermediate, malonyl-CoA (mal-CoA) and butyryl-CoA (but-CoA), intermediates of fatty acid synthesis and breakdown, respectively, as well as 3-hydroxy-3-methylglutaryl-CoA (hmg-CoA), an intermediate in the mevalonate and ketogenesis pathways. Due to their occurrence in a large number of metabolic pathways, the relative distribution of cellular acyl-CoAs represents a fingerprint of cell metabolism that may provide information on the activity of metabolic or signaling pathways including epigenetic regulations. Thus, analytics of acyl-CoAs has become a prime focus of interest for current metabolomics [4,5] and epigenomics [6]. While spectrophotometric [7] and fluorescence-based assays [8,9] are rather indirect measures and are limited to acetyl-CoA, more species can be quantified by classical high-pressure liquid chromatography (HPLC) [10,11,12]. More recently, HPLC coupled to tandem mass spectrometry (LC-MS/MS) has been applied to resolve a larger number of acyl-CoAs [13,14,15,16,17,18], and advanced methods based on stable isotope labelling have been recently proposed for acyl-CoA quantification in cellular models [19]. Nuclear magnetic resonance (NMR) may still represent another means for more rapid and less preparation-intensive quantification of highly abundant acyl-CoAs [20].

Owing to the large divergence in their acyl chains, acyl-CoAs are a very heterogeneous group of metabolites. They include hydrophobic and hydrophilic neutral or charged species of different carbon chain length, saturation and branching levels. In addition, their cellular concentrations differ by orders of magnitude, ranging from about 100 µM (for acetyl-CoA) to sub-µM levels [12,14,19,21]. It is therefore difficult to analyze the entire acyl-CoA metabolome with a single analytical approach. 

Here, we tested and compared simple and more advanced techniques with different extraction protocols to detect mainly short-chain acyl-CoAs from liver tissue. These include HPLC and a newly developed LC-MS/MS protocol, together with fluorimetric, spectrophotometric and NMR assays for acetyl-CoA. We challenge these techniques for detecting ischemia-induced changes in the acyl-CoA metabolome of rat liver. Research into this issue has mainly focused on acetyl-CoA and yielded largely divergent results in respect to timing and the amplitude of the changes. In rodent models, there are reports on the almost total depletion of hepatic acetyl-CoAs within 60 s of ischemia [22], while others observed a more moderate and much slower decrease [17,23]. Here, we applied our multi-method acyl-CoA metabolomics to two scenarios of hepatic ischemia, distinguished by their effect on cellular ADP/ATP and AMP/ATP ratios: mild (1–2 min) and advanced ischemia (5–6 min). Mild ischemia with a moderate change in adenylate ratios strongly decreases succinyl-CoA, but largely preserves acetyl-CoA and even increases some other acyl-CoAs upstream of the TCA cycle. Only advanced ischemia, characterized by more drastically altered adenylate ratios, also decreases acetyl-CoA, albeit still less pronounced than reported earlier [22]. Collectively, we compare a set of analytical approaches, with LC-MS/MS providing the most complete coverage of short-chain acyl-CoA metabolome for non-biased analysis.

## 2. Results

### 2.1. Ischemia Model and Metabolite Extraction

The lack of oxygen and nutrient supply during ischemia gradually affects different metabolic pathways. We choose to analyze two different conditions in rat liver, namely mild (1–2 min) and advanced ischemia (5–6 min), to capture not only the extent of the changes in the acyl-CoA metabolome, but also their timing,

Ischemia was induced ex vivo by delayed freeze-clamping of excised liver samples kept at room temperature. Freeze-clamping is essential for the immediate quenching of the metabolism to achieve metabolite extraction levels that are close to the physiological state. Protein denaturation and fast extraction from frozen tissue powder is then ensured either by perchloric acid (PCA) to yield aqueous extracts used for HPLC, fluorimetric and spectrophotometric assays, or by an organic–aqueous mixture (acetonitrile/methanol/water 2:2:1 *v*/*v*/*v*) used for NMR and LC-MS/MS. Aqueous extracts restrict analysis to hydrophilic short-chain acyl-CoAs, while the organic–aqueous mixture extracts more hydrophobic acyl-CoAs, but requires a lyophilization–resolubilization step. The techniques described in the following sections use livers from different ischemia experiments that were carried out at different periods of time.

### 2.2. HPLC Identifies CoA, Succinyl-CoA and Acetyl-CoA in Control and Ischemic Rat Liver 

First, we set up an HPLC protocol as our reference method, which provides a good compromise between rapidity, precision and coverage of CoA, acetyl-CoA and succinyl-CoA (Figure 1A,B for liver extracts and standards, respectively). The corresponding peaks were verified by spiking and hydrolysis experiments. Since the peaks were unchanged in the presence of dithiothreitol (DTT), we concluded that our extraction did not cause the formation of CoA disulfides (Figure 1A,B). HPLC could detect differences in all three CoA species between the control and ischemic samples (Figure 1C). The immediate processing of samples preserved the endogenous acyl-CoA levels as shown by a full recovery of internal standards (Table 1). In the same extracts, adenine nucleotides were determined as a direct, quantitative readout of ischemia (Figure 1D).

### 2.3. HPLC-Based Quantification of Acetyl-CoA, Succinyl-CoA and CoA Concentrations in Mild and Advanced Rat Liver Ischemia

In liver samples freeze-clamped 1–2 min after excision, we observed only moderate ischemia as judged by a limited decrease in ATP (≈30%) relative to controls (Figure 2B). Under these conditions, there was a clear drop in succinyl-CoA (≈50%), while values for acetyl-CoA and total CoA remained unchanged (Figure. 2A). We then freeze-clamped liver samples 5–6 min after excision to obtain more advanced ischemic conditions with a strong decrease in ATP (>60%; Figure 3B) relative to the controls and a doubling of the ADP/ATP ratio relative as compared to mild ischemia (Figure 2C and Figure 3C). In these advanced ischemic livers, in addition to decreased succinyl-CoA as in mild ischemia, we also observed reduced acetyl-CoA levels (≈40%; Figure 3A). Total free CoA increased, indicating an enhanced general hydrolysis of CoA esters. Our findings are further supported by plotting concentrations of CoA species directly versus ATP as a measure for the severity of ischemia. While succinyl-CoA decreases and CoA increases with a drop in ATP (Figure 4A,B), this relationship is non-linear for acetyl-CoA, which decreases only with a larger drop in the ATP concentration (Figure 4C). 

### 2.4. Mass Spectrometry to Quantify Acetyl- and Other Acyl-CoAs in Mild Liver Ischemia

For an extended coverage of acyl-CoA species, LC-MS/MS is the method of choice. Different protocols with variable complexity have been published during recent years, but this approach has not yet become standard in the field, likely due to its high technicity and costs. Here, we set up a simple method based on a single extraction with acetonitrile/methanol/water 2:2:1 *v*/*v*/*v*, which may not quantitatively extract all the acyl-CoA species that we detected with standards (Appendix A). LC-MS/MS allowed for a quantification of most of the 16 analyzed species, with the marked limitation of very hydrophobic, long-chain acyl-CoA species (Appendix A). LC-MS/MS also yielded by far the lowest limit of detection compared to all the methods applied here (in the low fmole range), satisfying linearity over several orders of magnitude (Table 1, Appendix A).

LC-MSMS confirmed our HPLC data, namely that mild ischemia did not immediately decrease acetyl-CoA levels in rat liver but rather led to a pronounced drop in succinyl-CoA (Figure 5). Also, 3-hydroxy-3-methylglutaryl-CoA, which is an intermediate rather involved in anabolic pathways like ketogenesis and the mevalonate pathway, tended to decrease (*p* = 0.08). By contrast, acyl-CoAs like propionyl-, butyryl-CoA or methylcrotonyl-CoA (mcro-CoA), which are intermediates of beta-oxidation and branched amino acid catabolism, and eventually feed into the TCA cycle, increased or showed a tendency to increase (*p* = 0.1) (Figure 5).

### 2.5. NMR to Quantify Acetyl-CoA and CoA in Mild Rat Liver Ischemia

Next, we aimed at using NMR as an alternative approach that allows for a direct metabolite quantification in an extract without the bias introduced by further processing with chromatography- and MS-based techniques. However, only abundant metabolites are amenable to NMR analysis, in our case acetyl-CoA and CoA (Figure 6). The NMR of liver extracts from mildly ischemic liver revealed a slight but significant decrease in acetyl-CoA not observed before, suggesting that this technique may have an advantage in terms of precision (Figure 6). However, this decrease was minor (<20%) compared to changes observed with other acyl-CoAs by HPLC and MS/MS. CoA levels showed an equivalent increase.

### 2.6. Fluorimetric and Spectrophotometric Assays to Quantify Acetyl-CoA in Rat Liver Ischemia

Finally, we applied photo- and fluorimetric assays to assess their performances in comparison with our more advanced analytics. These assays can be applied in any lab equipped with spectro- or fluorimeters, but they are limited to the determination of acetyl-CoA.

A classical spectrophotometric determination of acetyl-CoA involves a coupled enzymatic assay [24]. This two-step procedure requires careful setup and calibration. Although it is the least sensitive amongst our assays, its methodological variability and recovery rates approach the HPLC method (Table 1, Appendix A). Like with HPLC and MS/MS, no decrease in acetyl-CoA was determined in mildly ischemic liver. However, while HPLC determined a drop in acetyl-CoA in advanced liver ischemia, the spectroscopic test failed to show a significant decrease (representative experiments shown in Figure 7). Thus, either this test is less sensitive, or ischemia generates metabolites that interfere with the coupled enzymatic reaction applied.

Second, we used a fluorimetric assay proposed by a commercial supplier (Sigma, MAK039, Sigma-Aldrich, St. Louis, MO, USA). This multi-step procedure includes free CoA removal, conversion of acetyl-CoA into free CoA and CoA quantification with an NADH-generating reaction that leads to the reduction and thus fluorescence in a specific probe. Although the theoretical limit of detection and the methodological variability of this assay were acceptable, we observed variability in recovery and in-between experiments (Table 1, Appendix A). Still, this method confirmed that acetyl-CoA levels do not change significantly during mild ischemia, while there was a tendency of the levels to decrease in advanced ischemia (representative experiments shown in Figure 8).

## 3. Discussion

Despite being phylogenetically ancient and metabolically central intermediates, the cellular acyl-CoA levels and their (patho)physiological changes remain rather uncertain. Some of the variability in short-chain acyl-CoA concentrations present in the literature may be due to the methodology used, from extraction to analytics. Here, we directly compared for the first time several parallel methods able to determine hepatic acyl-CoAs and applied them to analyze ischemia-induced changes. Using rapid shock-freezing and extraction, we observed consistent results across different analytical approaches. Valid estimates of CoA and acetyl-CoA could already be obtained by simpler fluorimetric or spectrophotometric methods. An interesting alternative for these two metabolites is NMR, which combines exact quantification with minimal sample preparation and preservation of the sample for further analysis. The basic protocols applied here for HPLC and MS/MS extend the analysis to a variety of short- and medium-chain acyl-CoAs species. LC-MS/MS is definitively the most sensitive technique. It could also be calibrated by isotope spiking to obtain absolute quantities. With HPLC, NMR and spectrophotometric assays, we consistently determined about 50–100 nmol/g wet weight CoA and acetyl-CoA. This corresponds to the upper limits of what is reported for absolute values in the literature [11,13,14,18,21,25], also arguing for an excellent preservation of acyl-CoA species by our approaches. Other short-chain acyl-CoAs, except succinyl- and butyryl-CoA, were much less abundant. Indeed, some of them are close or even below the detection limits of LC-MS/MS.

There is ample evidence that hepatic ischemia can affect acyl-CoA levels, but there is just as much disagreement about the degree and timing of these events. In particular for acetyl-CoA, there is a large discrepancy among the reported values for the early ischemic phase, ranging from no or moderate changes [17,23] to almost full depletion [22]. According to our data, acetyl-CoA levels do not decrease during an early ischemic phase, similar to short ischemia in the brain [26] and heart [27,28]. We further observed that sustained levels of acetyl-CoA correlate with only moderately decreased hepatic energy parameters. In contrast, persisting liver ischemia that strongly impaired cell energetics also decreased acetyl-CoA levels, yet without being almost depleted as reported earlier [29]. Such a correlation between cell energetics and acetyl-CoA levels was also observed for cardiac ischemia [27]. Thus, the severity of ischemia as given by the cellular energy state, at least up to 6 min of ischemia, seems to be a good predictor for acetyl-CoA steady-state levels.

Unlike acetyl-CoA, most of the other short-chain acyl-CoAs showed rapid changes already upon mild ischemia (Figure 9). Both 3-hydroxy-3-methylglutaryl-CoA and succinyl-/methylmalonyl-CoA were depleted, while butyryl-CoA accumulated, and propionyl- and methylcrotonyl-CoA tended to increase. Indeed, decreased succinyl-CoA and increased propionyl-CoA levels emerge as hallmarks of ischemia across different species and tissues, including the liver, heart and brain [17,27,30]. In particular, the ischemic heart shows a very similar pattern of altered short-chain acyl-CoAs, with decreasing succinyl- and methylmalonyl-CoA and increasing butyryl- and propionyl-CoA [27]. As for acetyl-CoA, in contrast to liver, cardiac ischemia leads even to an initially strong increase [27,28]. Beyond ischemia, reduced succinyl-CoA levels were observed in multiple stress situations that impair cell energetics, including chronic heart failure [31] and chemical cell transfection methods (own unpublished data).

Collectively, our data on mild ischemia fit to a model wherein intermediates of catabolic pathways upstream of acetyl-CoA accumulate (methylcrotonyl-, butyryl- and propionyl-CoA), while certain intermediates of the TCA cycle (succinyl-CoA) or downstream anabolic pathways (3-hydroxy-3-methylglutaryl-CoA) deplete (Figure 9). These opposite effects on the acyl-CoA profile could be explained by considering the known impact of ischemia on the corresponding metabolic pathways. Ischemia rapidly inhibits mitochondrial respiration, which leads to an accumulation of ADP, NADH and FADH in the mitochondrial matrix [32]. This triggers the feedback inhibition of TCA cycle enzymes upstream of succinyl- and 3-hydroxy-3-methylglutaryl-CoA synthesis [33,34], contributing to a depletion of these two CoA species (Figure 9). In addition, under ischemia, the available succinyl-CoA is more rapidly converted into succinate to generate high energy phosphates (GTPs) [30]. Since further downstream the succinate dehydrogenase reaction is inhibited [30] or even reversed [35,36] in ischemia, succinate appears as a dead-end metabolite that cannot be further metabolized. It rapidly accumulates during ischemia across different tissues and species, with protective effects during the ischemic period, but detrimental consequences during reperfusion [37].

By inhibiting acetyl-CoA entry into the TCA cycle, and with nutrients not yet exhausted, ischemia is initially not accompanied by a marked loss of steady-state acetyl-CoA levels, at least in our model of mild hepatic ischemia and as reported by many other groups [17,27,28]. A detectable decrease only occurs in longer-lasting, advanced ischemia (e.g., >5 min). Possibly, the decisive factor here is the availability of external and cellular nutrient sources, which may differ among models. Our data, however, may well describe the in vivo ischemic situation, where the metabolization of available nutrients will delay acetyl-CoA depletion. Finally, TCA cycle inhibition may also be a causal factor for the increase in acyl-CoAs upstream of the cycle, in particular in the case of methylcrotonyl- and propionyl-CoA, two intermediates of branched-chain amino acid catabolism.

With respect to the methodologies compared in this study, all techniques except NMR agreed that mild hepatic ischemia does not significantly decrease acetyl-CoA levels. The significant but rather small (<20%) decrease reported by NMR was mainly due to very low data variability. Where measured, the methods also agreed on a more prominent decrease in acetyl-CoA levels (about 40%) in advanced ischemia (HPLC, fluorimetric tests) and a strong depletion of succinyl-CoA already in mild ischemia (HPLC, LC-MS/MS). Accordingly, CoA levels increased where measured (HPLC, NMR), only slightly in mild ischemia, and in more pronounced manner in advanced ischemia. Absolute quantities of acetyl-/acyl-CoAs could be obtained relatively fast and robustly by HPLC and NMR, but detection limits restrain the number of detectable species. LC-MS/MS clearly provides the largest coverage of acyl-CoA species and the lowest detection limits, but it may be more technically demanding to obtain absolute concentrations. Easily accessible spectrophotometric and fluorimetric assays for acetyl-CoA can also provide valuable information, if higher detection limits, larger data variability and limited absolute quantification are acceptable. Furthermore, as seen in our case with the spectrophotometric assay, potential interfering compounds in the extract of a given tissue may perturb these measurements, and an independent, additional method should always be applied as a control.

## 4. Materials and Methods

### 4.1. Tissue Harvesting and Ischemia

Male Wistar rats (about 3 months old, Charles River) were housed in the animal facility at 22 °C and fed with standard chow (3430PMS10, Serlab, Montataire, France). Animals were fasted for 16 h prior to experimentation; however, ischemia effects on acetyl- and succinyl-CoA were not affected when this fasting period was not applied. Liver removal was performed in the stable, strictly controlled environment of the animal facility, where rats were maintained under controlled 12 h light/12 h dark cycles (7:00 am/7:00 pm) at a temperature of 24 ± 2 °C and 40–70% humidity, without further humidification, and using a cold optical fiber light source to avoid drying. Rats were anesthetized by 2% isoflurane, and the abdomen was opened with care in order not to damage the diaphragm. Two liver samples (control and ischemic) were taken from each animal. A control sample was freeze-clamped directly in situ or immediately after excision (as detailed in the figure legends). To induce ex vivo ischemia, a liver sample was excised and placed on a glass dish for the desired time period (1–2 or 5–6 min) before freeze-clamping. Tissue samples were immediately used for sample preparation or stored at −80 °C. 

### 4.2. Sample Preparation

All procedures were performed at 4 °C or using liquid nitrogen. Approximately 200 mg of frozen liver tissue was ground in liquid nitrogen and processed immediately (for an overview, see Figure 10). 

To extract water-soluble acyl-CoAs or adenine nucleotides (in short, aqueous extraction), freeze-clamped livers were homogenized in 0.5 M perchloric acid (2:1 *v*/*w*; approx. 400 µL) in liquid nitrogen for immediate quenching. After thawing at room temperature, 4 M perchloric acid (1:10 *v*/*w*) was added, and the homogenate was incubated on ice for 30 min with occasional mixing and centrifuged (3000× *g*, 12 min, 4 °C). Then, the supernatant containing the metabolites was neutralized by addition of 5 M K_2_CO_3_ and centrifuged again (3000× *g*, 12 min, 4 °C). The resulting supernatant was immediately used for analysis or stored at −80 °C. To estimate recovery rates, parallel samples were spiked with acetyl-CoA solution (55 nmol/g fresh weight; Sigma-Aldrich, USA) during tissue grinding as internal standard (Appendix A). This concentration was chosen to approximatively double the endogenous hepatic acetyl-CoA concentration. 

For extraction of a larger number of acyl-CoA species, a mixed organic–aqueous solvent was used, containing acetonitrile/methanol/water (2:2:1, *v*/*v*/*v*) according to a protocol described earlier [16] (in short, organic extraction). Briefly, a 20-fold excess (v/w; approx. 4 mL) of precooled extraction solution (−20 °C) was rapidly added to frozen liver tissue powder to quench enzymatic reactions. Once thawed, samples were homogenized on ice with an Ultra-Turrax (IKA Labortechnik, Germany) and incubated on ice for 30 min with manual mixing twice. After centrifugation at 3660 x g for 5 min at 4 °C (Avanti JS 4.0), the supernatant was immediately dried with a Speedvac (Genvac, York UK), and dried metabolites were immediately resolubilized for direct use in NMR or LC-MS/MS protocols (see below), or stored at −80 °C.

### 4.3. HPLC

CoA, acetyl- and succinyl-CoA, as well as adenine nucleotides, were determined in aqueous protein-free extracts using an HPLC (Varian 410, Grenoble, France) with an RP-C18 column (Polaris C18-A 4.6 × 250 mm, 5 μm, Varian, France; ref. A2000250R046) at a flow rate of 1 mL/min and a column temperature of 30 °C, as described in detail elsewhere [11,38]. Briefly, for adenine nucleotides, the protein-free extract (75 μL premixed with 50 μL mobile phase) was separated in pyrophosphate buffer (28 mM, pH 5.75) and elution followed by absorption at 254 nm. For CoA, succinyl-CoA and acetyl-CoA, the protein-free extract (75 μL premixed with 50 μL mobile phase) was complemented with DTT (2 mM final) and separated with a mobile phase consisting of 100 mM mono-sodium phosphate, 75 mM sodium acetate, pH 4.6 combined with acetonitrile in the proportion 94:6 (*v*/*v*) and elution detected by absorption at 259 nm. The elution peaks were integrated using the STAR software (Varian, Grenoble, France).

### 4.4. LC-MS/MS

All experiments were carried out on a Vanquish Flex Binary liquid chromatography (LC) system coupled to a Q Exactive Plus Orbitrap Mass spectrometer (MS) equipped with a HESI probe (Thermo Fisher Scientific, Waltham, MA, USA). Acyl-CoA stock solutions (15 mM) prepared in ammonium acetate (10 mM, pH 8)/methanol/water (2:2:1, *v*/*v*/*v*) were serially diluted to a range from 2.5 × 10^−4^ to 5 × 10^−11^ M. Dry pellet samples of tissue metabolites were solubilized in 40 µL of the same solution. Samples were then centrifuged at 14,000× *g* and 4 °C for 15 min, and supernatants (5 μL for standards and 10 μL for samples) injected into the LC-MS system. LC was performed on a Acquity HSS T3 column (150 × 2.1 mm i.d., 1.8 µm; Waters) using 5 mM ammonium acetate pH 8 in water (mobile phase A) or acetonitrile (mobile phase B) with a flow rate of 0.2 mL/min and a column temperature of 30 °C. The elution gradient was 5.2% phase B initially, which then increased to 21% over 10 min and to 100% over 5 min, kept at 100% B for 5 min, and finally reequilibrated to 5.2% and held for 4 min. Settings for MS in positive mode were as follows: heater temperature, 120 °C; sheath gas, 80; auxiliary gas, 12; spray voltage, 4 kV. Capillary temperature was set at 280 °C, and S-lens was 50. A full scan range was set from 100 to 1500 (*m*/*z*). For standards, analysis in Full MS1 and MS/MS was performed. For full MS1 analysis, the resolution was set at 70,000 (at *m*/*z* 200), the maximum injection time (max IT) was 200 ms and the Automated Gain Control (AGC) was targeted at 3 x 10^6^ ions. For targeted MS/MS analysis, the resolution and AGC were 35,000 and 200,000, respectively. The isolation width of the precursor ion was set at 1.5 (*m*/*z*), collision energy (CE) was 25 and max IT was 100 ms. For samples, PRM assays were performed using an inclusion list containing the m/z and RT of 16 acyl-CoAs; PRM consisted of targeted MS/MS scans in HCD mode. Acquisition was performed in positive ion mode with a resolution of 35000. Data were analyzed with different software (Thermo Fisher Scientific, Waltham, MA, USA). Chromatograms were analyzed with Excalibur (Qual browser program) and relative quantification of acyl-CoAs was performed with Trace finder. Concentration series of a mixture of acyl-CoA species were run to check for signal recovery (Appendix A) and linearity (Appendix A).

### 4.5. NMR

Dried organic extracts were resuspended in 650 μL of NMR deuterated buffer (sodium phosphate buffer 0.2 M in 100% D_2_O, pH = 7.4, with 0.002 mM NaN_3_) and vortexed for 45 s. Samples were then transferred into 1.5 mL Eppendorf tubes and centrifuged (5 min, 13,000 rpm, 8.3 cm rotor, 4 °C). A total of 550 μL of the supernatant was then transferred into a standard 5 mm NMR tube. NMR experiments were conducted on a Bruker 700 MHz (^1^H resonance frequency) NMR spectrometer, equipped with a triple-resonance TCI cryoprobe. The temperature was controlled at 27 °C throughout NMR acquisition. After automatic tuning and shimming, one-dimensional NOESY experiments with water pre-saturation (pulse program noesygppr1d with 10 ms NOESY mixing time) were recorded. The spectral width was set to 20 ppm and the acquisition time was 2 s with a relaxation delay of 3 s, corresponding to a total acquisition time of 21 min per spectrum (256 transients co-added). A 0.3 Hz exponential window functional was applied to the raw data prior Fourier transform. Spectra were manually phased, and baseline-corrected. Chemical shifts were referenced to the glucose doublet at 5.23 ppm. A 1.12 mM lactate external reference sample was used for absolute quantification from the NMR data using a synthetic reference signal inserted (ERETIC2 calibration method [39,40]). Quantification of CoA and Ac-CoA was then obtained by interactive spectrum deconvolution using the ChenomX NMR suite, based on the respective characteristic singlets observed for each compound around 8.54 ppm (Appendix A) and 0.71 ppm.

### 4.6. Spectrophotometric and Fluorimetric Assays

Acetyl-CoA was quantified in a microcuvette format by a spectrophotometric assay using a sequential, coupled enzymatic reaction as described earlier [24] with minor modifications. Mainly, the assay was downscaled to a final volume of 0.503 mL (instead of 2.01 mL) to reduce consumption of metabolite extract. Furthermore, for a most exact determination, the amount of acetyl-CoA in the assay should be kept below 50 nmol. Alternatively, acetyl-CoA was quantified in a 96-well microplate format by a commercial fluorescence assay kit (MAK039, Sigma Aldrich, USA) according to the supplier’s instructions. Deviating from this protocol, the entire fluorescence kinetics at Ex/Em = 535/587 nm was recorded for up to 30 min for retroactive choice of the endpoint.

### 4.7. Calculations and Statistics

Recovery of acetyl-CoA was calculated as follows: R = spiking added (nmol/g)/spiking recovered (nmol/g). Detection limits of each method were estimated from the lowest concentration detectable in calibration curves after blank correction (see, for example, Appendix A). All data are expressed as mean ± SEM. Statistical analysis was performed using Sigma Plot 9.0. Differences were considered significant at a probability level of *p* < 0.05, if not stated otherwise.

## 5. Conclusions

Our data show that targeted acyl-CoA metabolomics can trace metabolic changes during ischemia or other physio-pathological conditions. However, the rapid quenching of metabolism and the choice of the best adapted method are crucial. While HPLC and NMR are methods of choice for rapid and quantitative analytics of specific acyl-CoAs, LC-MS/MS is essential for full-fledged acyl-CoA metabolomics. Ischemia, in general, is known to trigger a complex stress response leading to disruption in the cellular metabolic networks. We observed, in mild ischemia, rather preserved acetyl-CoA levels, with strong depletion only seen with succinyl-CoA. These data lend support to a hepatoprotective effect during early ischemia [41], where the metabolic response may allow for hepatocyte survival. Finally, altered steady-state levels of short-chain acyl-CoAs as determined here not only redirect anabolic and catabolic fluxes, but also reshape the landscape of secondary protein modifications, including epigenetically relevant histone acylations [2,42]).

## Figures and Tables

**Figure 1 ijms-24-14957-f001:**
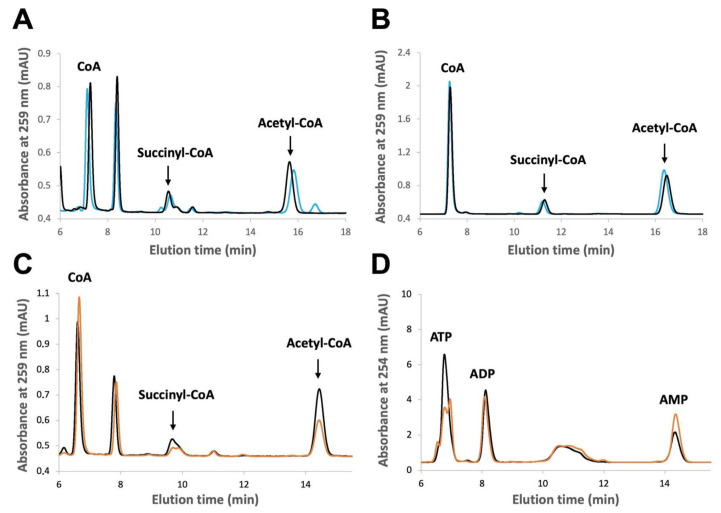
Representative HPLC traces for detection of CoA, acetyl-CoA, succinyl-CoA and adenylates. (**A**) CoA, succinyl-CoA and acetyl-CoA detected by HPLC in PCA extract of rat liver pretreated (blue) or not (black) with 2 mM DTT (representative tracings for ischemic liver, freeze-clamped after 1–2 min ischemia ex vivo). (**B**) Standards for CoA, succinyl-CoA and acetyl-CoA (from left to right) detected by HPLC, pretreated (blue) or not (black) with 2 mM DTT. (**C**) CoA species detected by HPLC in PCA extracts of control (black) and ischemic (orange) liver samples (either freeze-clamped immediately after excision or after 5–6 min ischemia ex vivo, respectively; representative tracings from the same animal). (**D**) ATP, ADP and AMP detected by HPLC in the samples analyzed in (**C**). A peak eluting after CoA in (**A**,**C**) does not correspond to any CoA species analyzed in this study. Rat livers were harvested after 16 h of fasting for the animals and immediately processed for PCA extraction. Representative traces from triplicates.

**Figure 2 ijms-24-14957-f002:**
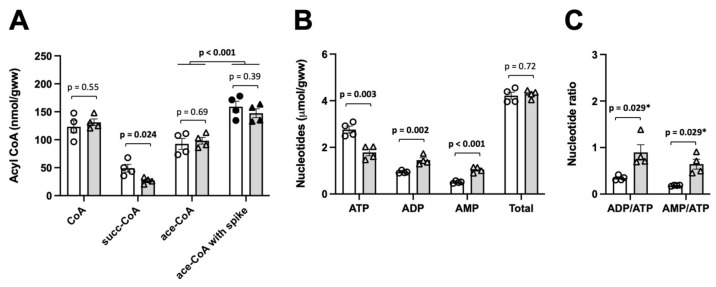
HPLC analysis of acetyl-CoA, succinyl-CoA and CoA in mild rat liver ischemia. (**A**) CoA species, (**B**) adenine nucleotides and (**C**) adenine nucleotide ratios measured by HPLC in PCA extracts. Control (white bars, circles) or mild ischemic liver samples (grey bars, triangles) were obtained from the same animal by freeze-clamping directly in situ or after 1–2 min ischemia ex vivo, respectively. Analysis was repeated with spiking for acetyl-CoA (55 nmol/g), revealing the absence of losses during our extraction and analysis procedure. Abbreviations: ace, acetyl; succ, succinyl. Note: adenine nucleotides give a direct measure for the severity of rat liver ischemia. Rat livers were harvested from 4 animals after 16 h of fasting and immediately processed for PCA extraction. Data are presented as mean ± SEM (n = 4) and analyzed by Student’s *t*-test (or *Mann–Whitney test for comparison between control and ischemia).

**Figure 3 ijms-24-14957-f003:**
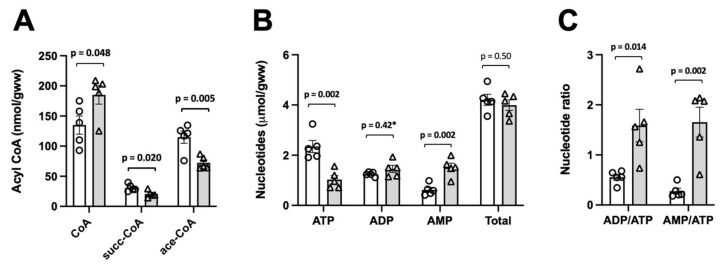
HPLC analysis of acetyl-CoA, succinyl-CoA and CoA in advanced rat liver ischemia. (**A**) CoA-species, (**B**) adenine nucleotides and (**C**) nucleotide ratios measured by HPLC in PCA extracts of control (white bars, circles) and advanced ischemic liver samples (grey bars, triangles) obtained from the same animal by freeze-clamping immediately after excision or after 5 min ischemia ex vivo. Livers were harvested from 5 animals after 16 h of fasting and immediately processed for PCA extraction. Random samples spiked with acetyl-CoA as in Figure 2 confirmed the absence of losses. Abbreviations: ace, acetyl; succ, succinyl. Data are presented as mean ± SEM (n = 5) and analyzed by Student’s *t*-test (or *Mann–Whitney test for comparison between control and ischemia).

**Figure 4 ijms-24-14957-f004:**
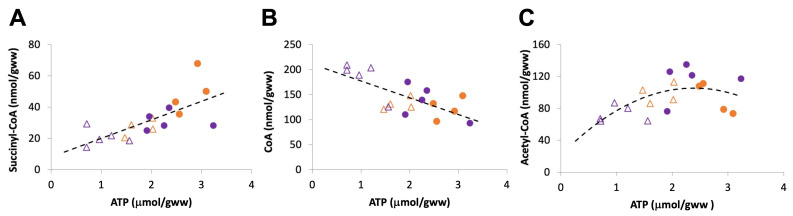
HPLC assays reveal relationship between severity of ischemia and levels of CoA species. HPLC data on CoA species in controls (closed circles) and ischemia (open triangles) are pooled and presented in dependence of ATP concentration as an indicator of the severity of ischemia. Colors identify the experimental series (orange, data from Figure 2; purple, data from Figure 3).

**Figure 5 ijms-24-14957-f005:**
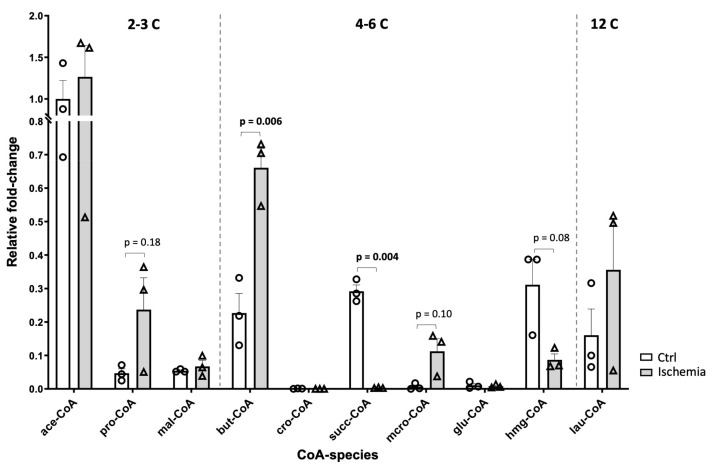
LC-MS/MS analysis of acyl-CoA species in mild rat liver ischemia. CoA species measured in organic extracts of control liver (white bars, open circles) and liver samples subjected to mild ischemia (grey bars, open triangles; see legend Figure 2). MS data were corrected for background and normalized to liver fresh weight. Relative abundance of acyl-CoA species is given after normalization to acetyl-CoA levels in controls. Data are mean ± SEM (n = 3 rats; each data point is a mean of 3 technical replicates), *p*-values of Students *t*-test are indicated for *p* < 0.2 and bold if significant. Non-detectable or non-quantifiable results are set to zero. Abbreviations: ace, acetyl (C2); pro, propionyl (C3); mal, malonyl (C3); but, isobutyryl and butyryl (C4); cro, crotonyl (C4); succ, succinyl and methylmalonyl (C4); isoval, isovaleryl (C5); mcro, methyl-crotonyl (C5); glu, glutaryl (C5); hmg, 3-hydroxy-3-methyl-glutaryl (C6); oct, octanoyl (C8); lau, lauroyl (C12).

**Figure 6 ijms-24-14957-f006:**
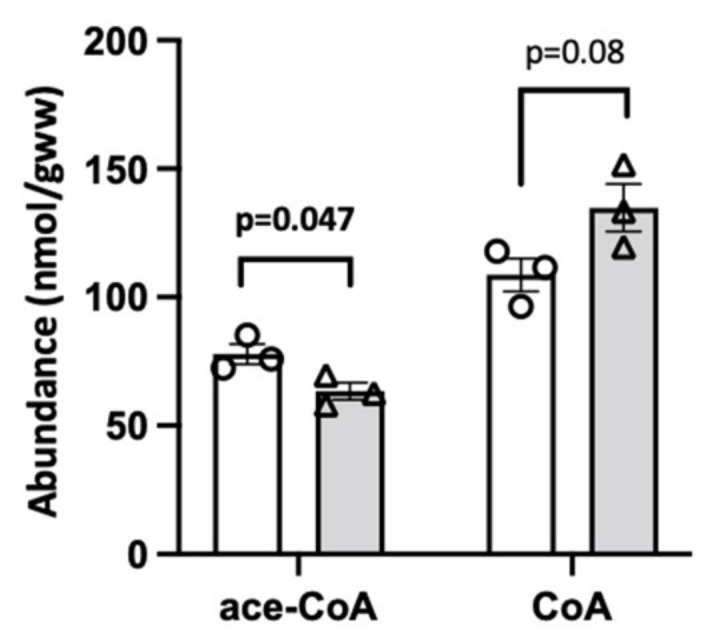
NMR analysis of CoA and acetyl-CoA in mild rat liver ischemia. CoA and acetyl-CoA measured in organic extracts of control liver (white bars, open circles) and liver samples subjected to mild ischemia (grey bars, open triangles; see legend Figure 2). Livers were harvested from 3 animals after 16 h of fasting and immediately extracted. Abbreviations: ace, acetyl. Data are presented as mean ± SEM (n = 3) and analyzed by Student’s *t*-test.

**Figure 7 ijms-24-14957-f007:**
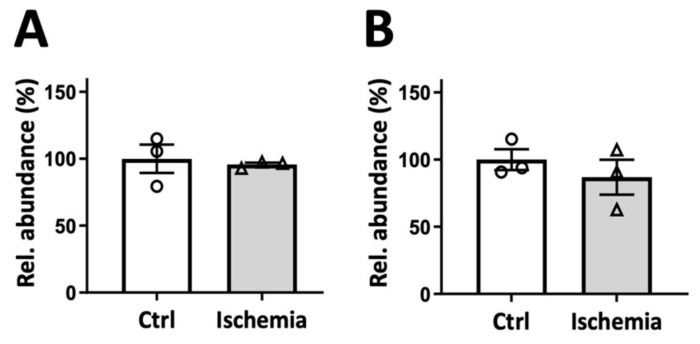
Spectrophotometric assay of acetyl-CoA in mild and advanced rat liver ischemia. Acetyl-CoA detected in PCA extracts of control liver (white bars, open circles) and liver subjected to ischemia (grey bars, open triangles), either (**A**) mild ischemia or (**B**) advanced ischemia (see legend Figure 2 and Figure 3). Livers were harvested from 3 animals after 16 h of fasting. Data are normalized to controls and presented as mean ± SEM (n = 3) and analyzed by paired Student’s *t*-test.

**Figure 8 ijms-24-14957-f008:**
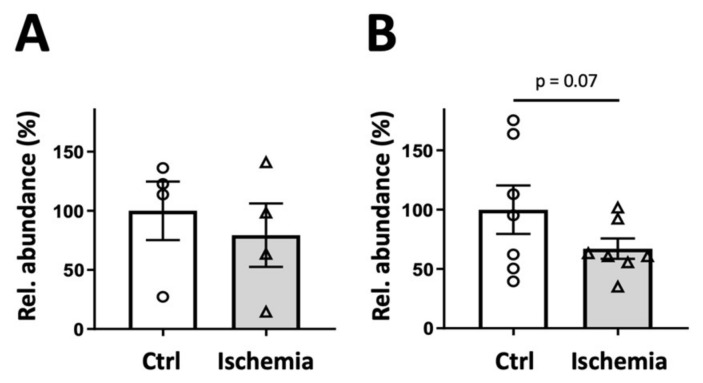
Fluorimetric assay of acetyl-CoA in mild and advanced rat liver ischemia. Acyl-CoA detected in PCA extracts of control liver (white bars, open circles) and liver subjected to ischemia (grey bars, open triangles), either (**A**) mild ischemia or (**B**) advanced ischemia (see legend Figure 2 and Figure 3). Livers were harvested from 4–7 animals after 16 h of fasting. Data are normalized to controls and presented as mean ± SEM (n = 4–7; data from 3 experiments) and analyzed by paired Student’s *t*-test.

**Figure 9 ijms-24-14957-f009:**
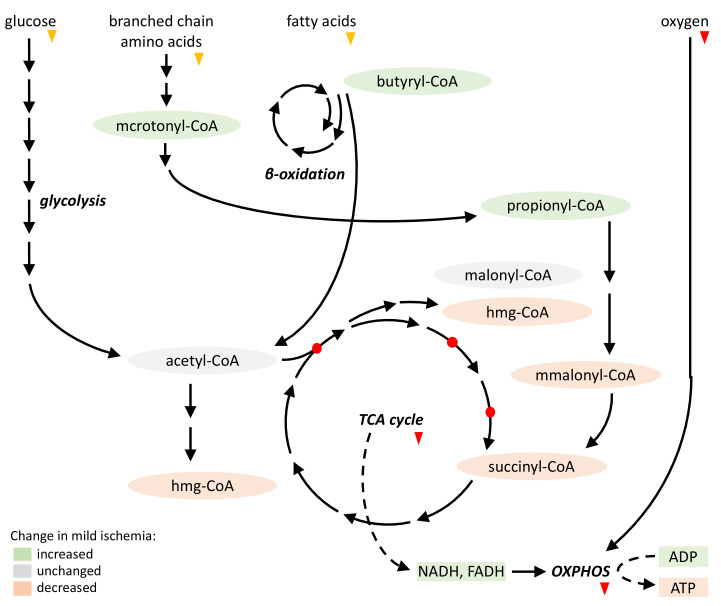
Detected acyl-CoAs and their position in major metabolic pathways. Scheme showing simplified metabolic pathways and their acyl-CoA intermediates as detected in this study by LC-MS/MS (see Figure 5). Changes (significant differences or trends) are color-coded. Lack of oxygen supply (red arrows) will halt oxidative phosphorylation (OXPHOS), leading to accumulation of ADP, NADH and FADH in the mitochondrial matrix. This in turn will inhibit TCA cycle enzymes (citrate synthase, α-ketoglutarate dehydrogenase, isocitrate dehydrogenase; red circles). Reduced external nutrient supply (orange arrows) may not yet be as important under these mild ischemic conditions. Abbr.: mcrotonyl, methylcrotonyl; hmg, 3-hydroxy-3-methyl-glutaryl: mmalonyl, methyl-malonyl.

**Figure 10 ijms-24-14957-f010:**
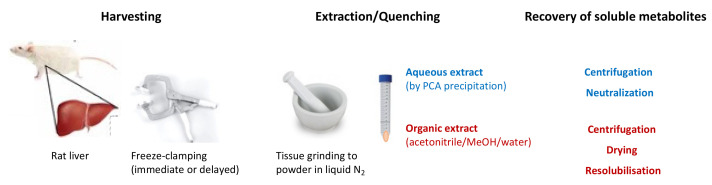
Schematic overview of sample preparation.

**Table 1 ijms-24-14957-t001:** Comparison of applied methods.

Method	Number of (Acyl-)CoAs Detected	Cost and Technicity	Accessibility	Limit of Detection ^(1) (2)^		Coefficient of Variation (%) ^(1) (7)^	Recovery Rate (%) ^(1) (10)^
				Quantity (Pmol)	Concentration (nM)		
Spectrophotometric assay	1	+	Laboratory	≈1000	≈2000 ^(3)^	11 ± 4 (n = 5)	72–96% (n = 6)
Fluorimetric assay	1	++	Laboratory	≈45	≈400 ^(4)^	23 ± 4 (n = 17)	23–61% (n = 6)
HPLC	3	++	Laboratory	≈7.5	≈250 ^(5)^	10 ± 5 (n = 3) ^(8)^	97–113% (n = 8)
NMR	2	+++	Facility	ND	ND	ND	ND
LC-MS/MS	12	++++	Facility	≈0.005	≈0.5 ^(6)^	5.4 ± 1.5 (n = 6) ^(9)^	ND

(1) Value for acetyl-CoA. (2) Lowest quantity or concentration detectable, derived from measured concentration series. (3) Final test volume: 500 µL. (4) Final test volume: 112 µL. (5) Sample volume injected: 30 µL. (6) Sample volume injected: 10 µL. (7) Repeated measurements of same sample. (8) Values of 11 ± 3 (n = 3) and 5 ± 2 (n = 3) for succinyl-CoA and CoA, respectively. (9) For other acyl-CoAs, see Appendix A. (10) Minimal and maximal recovery of internal standard (55 nmol/g acetyl-CoA) added before extraction. + to ++++, increasing costs and technicity; ND, not determined.

## Data Availability

Data are available upon reasonable request to the authors.

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
