# Peer review of "Multi-Method Quantification of Acetyl-Coenzyme A and Further Acyl-Coenzyme A Species in Normal and Ischemic Rat Liver"

_ijms, 2023, doi:10.3390/ijms241914957_

Round 1

Reviewer 1 Report

This study quantified cellular acyl-CoAs concentrations in normal and ischemic rat liver by using several methods. This study is well-written and well-developed, which only requires a minor revision before publication.

1.     A separate single Conclusion section is needed to emphasize the major results, significance, and novelty of this study. However, in this article, the conclusion is embedded into the Discussion section.

2.     The pronouns/names used in this article are not consistent. For example, in Figure 3 caption, acetyl-CoA and succinyl-CoA are used, while in Figure 3, ace-CoA and succ-CoA are used. This issue is also observed in Figure 4 – Figure 6. The authors should use one identical pronoun/name for a same material.

3.     Abbreviations should be provided immediately after the full name when it is first introduced, rather than being provided in the captions of the figures. Noted: the captions can still be preserved.

4.     Line 311: What are other mechanisms?

Author Response

Reviewer 1

This study quantified cellular acyl-CoAs concentrations in normal and ischemic rat liver by using several methods. This study is well-written and well-developed, which only requires a minor revision before publication.

We thank the reviewer for this very positive evaluation. We have revised/added text where necessary (marked in yellow).

  1.     A separate single Conclusion section is needed to emphasize the major results, significance, and novelty of this study. However, in this article, the conclusion is embedded into the Discussion section.

As suggested by the reviewer, the conclusion paragraph has been separated from the discussion as an independent chapter and was also extended (lines 460-481).

2.     The pronouns/names used in this article are not consistent. For example, in Figure 3 caption, acetyl-CoA and succinyl-CoA are used, while in Figure 3, ace-CoA and succ-CoA are used. This issue is also observed in Figure 4 – Figure 6. The authors should use one identical pronoun/name for a same material.

We thank the reviewer for having identified this inconsistency. We have added a description of abbreviations to Figures 2, 3 and 6 in the captions. As far as we see, Figure 4 does not include abbreviations for CoA-species, and Figure 5 already contains such a list. For best readability, we would like to keep the full names for acyl-CoAs in the text (but added abbreviations on first occurrence), and use the abbreviations only in the figures (where they are now fully explained again).

  1.     Abbreviations should be provided immediately after the full name when it is first introduced, rather than being provided in the captions of the figures. Noted: the captions can still be preserved.

We have added abbreviations after first appearance of the full name, but prefer to use them only in the figures and full names in text and tables (as mentioned above). We have verified that this is consistent now in the entire text.

  1.     Line 311: What are other mechanisms?

    There seem to be different metabolic pathways that lead to accumulation of succinate in ischemia as a sort of dead-end metabolite that cannot be further metabolized. In addition to the mentioned acceleration of succinyl-CoA conversion into succinate (powered largely by glutaminolysis, not glucose; Zhang et al 2018, Nature Metab), also downstream utilization of succinate by succinate dehydrogenase (Complex II) is inhibited {Zhang et al 2018, Nature Metab) or even reversed (Chouchani et al 2014 Nature; Chouchani et al 2016 Cell Metab). We have added the relevant information and references in lines 306-310.

Reviewer 2 Report

The paper "Multi-method quantification of acetyl-CoA and further acyl-CoA species in normal and ischemic rat liver" aims to compare different approaches to quantify cellular concentrations of Acyl-(CoA) carrying different acyl chains (acyl-CoAs), as the central intermediates of many metabolic pathways and donor molecules for protein lysine acylation. Assays are performed to compared data from rat liver specimens under normal and ischemic conditions. The results are valuable to indicate the potential of the different methods, but the text suffers from some drawback as detailed below. Thus, the Authors are invited to revise the text and resubmit.

In general, ischemia is described to be procured by oxygen deprivation of liver specimens by keeping them at room temperature after excision. It is to be reported the temperature and if the specimens were kept in a humidified environment, preventing drying even if for a limited time.

Results – the different techniques provide quantification of different kinds of Aacy-CoA, and LC-MS/MS gives the most numerous.  For the promise of the title and of the abstract, it should be expected first of all a more detailed comparison between the data provided from the different methods. Limiting to the molecular species detected by the different methods, ace-CoA for example, is seems that in general schemia results in no change or slight decrease of ace-CoA, while an increase is indicated by LC-MS/MS in mild ischemia, and a significant decrease is shown by HPLC in advanced ischemia. CoA, in turn, is shown to increase both by NMR and HPLC. These evidences should be duly considered along with the ability of LC-MS/MS to detect much more species.

The scheme in Figure 9 should be moved to the Introduction, as a preliminary illustration of the acyl-CoAs role in major metabolic pathways.

Figures 2,3- AceCo is there a reason for the absence of data for ace-CoA with spike in figure 3?

 Minor remarks

Bar charts in the related figures should be shown in the same style, in particular Fig. 5.

good

Author Response

Reviewer 2

The paper "Multi-method quantification of acetyl-CoA and further acyl-CoA species in normal and ischemic rat liver" aims to compare different approaches to quantify cellular concentrations of Acyl-(CoA) carrying different acyl chains (acyl-CoAs), as the central intermediates of many metabolic pathways and donor molecules for protein lysine acylation. Assays are performed to compared data from rat liver specimens under normal and ischemic conditions. The results are valuable to indicate the potential of the different methods, but the text suffers from some drawback as detailed below. Thus, the Authors are invited to revise the text and resubmit. 

We thank the reviewer for the overall positive evaluation, and we have done some revisions in the text to clarify the below mentioned issues (marked in yellow).

In general, ischemia is described to be procured by oxygen deprivation of liver specimens by keeping them at room temperature after excision. It is to be reported the temperature and if the specimens were kept in a humidified environment, preventing drying even if for a limited time.

The removal of liver was performed in the stable, strictly controlled environment of our animal facility, with limited variations of temperature (22°C +/- 2°C ) and humidity (40-70%). There was also no additional local heating by illumination, since we a used a cold light source based on optical fibers. Our fresh weight records do not show any (even minor) differences between liver freeze-clamped in situ and liver maintained on a glass dish for up to 6 min. We are therefore confident that drying of tissue was not an issue during these experiments, and we did not take additional humidification measures. We added relevant information on animal maintenance to the Methods section (lines 352-356).

Results – the different techniques provide quantification of different kinds of Aacy-CoA, and LC-MS/MS gives the most numerous.  For the promise of the title and of the abstract, it should be expected first of all a more detailed comparison between the data provided from the different methods. Limiting to the molecular species detected by the different methods, ace-CoA for example, is seems that in general schemia results in no change or slight decrease of ace-CoA, while an increase is indicated by LC-MS/MS in mild ischemia, and a significant decrease is shown by HPLC in advanced ischemia. CoA, in turn, is shown to increase both by NMR and HPLC. These evidences should be duly considered along with the ability of LC-MS/MS to detect much more species.

To clarify the results in respect to the different methods applied, we added some more interpretation to the discussion section (lines 330-346). In fact, all methods except NMR agreed that mild ischemia does not significantly decrease acetyl-CoA levels. Only NMR reported a significant (p=0,047) but small (<20%) decrease, mainly due to a low variability between samples. Where measured, advanced ischemia decreased acetyl-CoA levels by about 40% (HPLC and fluorimetric test; the spectrophotometric results likely being biased). Different methods (HPLC and MS/MS) also agreed in the fact that the Krebs cycle succinyl-CoA is already strongly depleted in mild ischemia. Accordingly, CoA levels increased where measured (HPLC, NMR), only slightly in mild ischemia, and more pronounced in advanced ischemia. Thus, HPLC and NMR seem to give a fast and quantitative access to acetyl-CoA levels, while LC-MS/MS has clearly the largest coverage but requires considerable input, in particular when aiming at absolute concentrations. Easily accessible photometric and fluorimetric methods can provide valuable information, but potential interfering compounds in the extract of a given tissue may require an independent method as a control. 

The scheme in Figure 9 should be moved to the Introduction, as a preliminary illustration of the acyl-CoAs role in major metabolic pathways.

Figure 9 is a highly simplified metabolic scheme that was designed to summarize our results. For example, only the acyl-CoA species detected in our study are mentioned, and the color codes indicate the observed changes. In our understanding, this is not a non-biased overview of metabolic pathways suitable for introduction, and an anticipated, detailed presentation of results may be even confusing. We therefore left the figure at its place in the discussion section.

Figures 2,3- AceCo is there a reason for the absence of data for ace-CoA with spike in figure 3?

Since our initial experimental series with mild ischemia and spiking revealed that our analytics does not lead to significant loss of spiked acetyl-CoA, we did not systematically include such spiking in all follow-up experiments, but only checked random samples. All measurements confirmed a preservation of the spiked acetyl-CoA. 

Minor remarks: Bar charts in the related figures should be shown in the same style, in particular Fig. 5.

Figure 5 has been revised to exactly correspond to the style used in other figures.

Round 2

Reviewer 2 Report

The paper "Multi-method quantification of acetyl-CoA and further acyl-CoA species in normal and ischemic rat liver" has been duly revised with careful attention to the comments of the Reviewer.  It is particularly appreciated the effort in providing a summarizing text on the results obtained (lines 379- 395) very helpful in facilitating the reader to understand and move along the whole work. Only one minor remark it about the absence of data for ace-CoA with spike in figure 3, and related answer. Since a reader could have the same question, at his convenience a sentence as provided in the related answer is to be added in the text. After that, the paper will be suitable for publication. 

good

Author Response

The paper "Multi-method quantification of acetyl-CoA and further acyl-CoA species in normal and ischemic rat liver" has been duly revised with careful attention to the comments of the Reviewer.  It is particularly appreciated the effort in providing a summarizing text on the results obtained (lines 379- 395) very helpful in facilitating the reader to understand and move along the whole work. Only one minor remark it about the absence of data for ace-CoA with spike in figure 3, and related answer. Since a reader could have the same question, at his convenience a sentence as provided in the related answer is to be added in the text. After that, the paper will be suitable for publication. 

We have added a sentence to the legends of Figures 2 (line 233) and 3 (line 244) to make clear that  Figure 2 already shows the absence of losses with your procedure, and that for Figure 3 only random samples were checked which confirmed the absence of losses.